# Effect of Antibacterial Peptide Microsphere Coating on the Microbial and Physicochemical Characteristics of *Tricholoma matsutake* during Cold Storage

**DOI:** 10.3390/polym14010208

**Published:** 2022-01-05

**Authors:** Hongli Li, Yan Feng, Peng Zhang, Mingwei Yuan, Minglong Yuan

**Affiliations:** National and Local Joint Engineering Research Center for Green Preparation Technology of Biobased Materials, Yunnan Minzu University, Kunming 650500, China; honglili@vip.163.com (H.L.); fy971129@163.com (Y.F.); newworldopen@163.com (P.Z.)

**Keywords:** antimicrobial peptide (microspheres), *Tricholoma matsutake*, physicochemical quality, microbial quality

## Abstract

The effect of novel antimicrobial peptides (AMPs) and antimicrobial peptide microspheres (AMS) on the physicochemical and microbial quality of *Tricholoma matsutake* wild edible mushrooms was investigated. In the experiments, 1.0 g/L, 0.5 g/L of AMS, and 1.0 g/L AMPs were used as preservatives. Mushrooms coated with 1.0 g/L and 0.5 g/L of AMS as a preservative had better physicochemical and sensory qualities than did mushrooms coated with 1.0 g/L of AMPs. In the experiment, 1.0 g/L of blank microspheres without cathelicidin-BF-30 (PLGA-1.0) and distilled water was used as the control. Samples with these two treatments had minimal changes in texture, weight loss, total bacteria count, and sensory attributes. Research results suggests that the use of AMS can maintain the quality of *Tricholoma matsutake* wild edible mushrooms and could extend the postharvest life to 20 d.

## 1. Introduction

Mushrooms have been widely used since ancient times as food and for medicinal or other functional purposes. In recent years, there has been an increase in the worldwide consumption of fresh wild mushrooms (e.g., *Lactarius deliciosus Boletus edulis Cantharellus* spp. *Hygrophorus* spp. and *Tricholoma matsutake* spp.) because of their delicate flavor, trace minerals, and texture [1]. *Tricholoma matsutake* is a fungus that belongs to the subgenus *Tricholoma*, and it is widely distributed in Asian countries, such as China, Korea, and Japan [2,3]. A large number of bioactive substances extracted from the fruiting bodies of *T. matsutake*, such as volatile compounds [4], polysaccharides [5], and polysaccharide–protein complex fractions [6,7], have been found to have immunomodulating and antioxidant properties [8,9]. *Tricholoma matsutake* is one of more precious and relatively expensive species throughout the world, exhibiting a characteristic and delicate flavor as well as several biological activities, such as sterol-lowering, anti-oxidant, immunomodulating, and anti-tumor effects in humans.

However, mushrooms perish rapidly and have a limited shelf-life of only 1 to 3 days at room temperature [10]. The high respiration rate, lack of physical protection to avoid water loss, and changes from microbial attack are often associated with a rapid decrease in mushroom quality [11]. An extended shelf-life is a key factor for making any food commodities more profitable and commercially available for longer periods of time at the best possible quality. The producer will benefit from a longer shelf-life by being able to sell the product in markets over greater distances [12]. There is a general trend in mushroom preservation research toward the development of preservation techniques that are practical and operable in order to reduce damaging to food products. These damages include mushroom loss of quality, contributing to their deterioration through browning, cap opening, stipe elongation, cap diameter increase, weight loss, and texture changes [13].

According to several authors, a coating could delay mushroom spoilage with minimum changes to the physiochemical and sensory quality of the mushrooms [14,15,16]. Edible coatings can maintain the quality of fruits and vegetables by functioning as solute, vapor, and gas barriers. These coatings could help decrease moisture loss and slow respiration by reducing fruit oxygen uptake from the environment [17]. Tiwari outlined the various types of animal and plant microorganisms isolated from antimicrobial applications in the food industry [18]. Edible coatings have been widely used to extend the shelf-life of various agricultural products [19]. Moreover, the product can be protected from microbial and mechanical damage, maintain a beautiful appearance, and prevent the escape of volatile substances [20]. Edible coatings derived from biodegradable and biomass can better meet the needs of customers and environmental package [21]. Coating preservation technology has focused on surface coating technology and the different types of coatings that are used to protect food from the effects of environmental factors, such as light, oxygen, and microorganisms [22]. Some summaries have been published on the effects of keeping perishables fresh by using coating technology, along with the shortcomings, future development trends, and applications. Research has found that the bacteriostatic action of antimicrobial peptides is achieved by destroying bacterial cell membranes and playing a role in antibacterial mechanisms, which is apparently not easy in the creation of medicines [23].

In recent years, antimicrobial peptides (AMPs) have been used as potential new antibiotics to solve the problem of resistance, acting as a kind of biological epidemic prevention system against exogenous pathogens of natural antibacterial peptides, which are small molecules composed of 12–100 amino acids [24,25]. Once in a target microbial membrane, the peptide kills target cells through diverse mechanisms [26]. Thus, antimicrobial peptides are widely applied in the fields of medicine, agriculture, aquaculture, and food industry because of their small molecular weight [27,28,29]. Although antibacterial peptides may have unprecedented application prospects for edible fungus such as precious wild fungus—as an antistaling agent in particular—antibacterial peptides have not yet been used in the preservation of wild fungus.

Nevertheless, due to its short half-life, AMPs can only be stable for 2.5 h in rat plasma [30]. Embedding it into a polymer microsphere material can protect it from the effects of the external environment and improve its half-life. Polylactic acid polymer microsphere materials play an important role in the field of drug-controlled release. In recent years, the structure has become more complex, and new materials have shown excellent performance. There is increasing interest in using PLA for food packaging applications. PLA is a biodegradable polymer that can be produced from annually renewable resources, such as sugar beet or corn starch [11]. Additionally, combinations of natural polymer materials with synthetic polymer materials integrate the advantages of both, with polypeptides, proteins, nucleic acids, and other new drugs employing controlled release applications.

In this work, the antimicrobial peptide microspheres and antimicrobial peptides as preservatives were carried out for the preservation experiments of Tricholoma matsutake. The purpose is to test the fresh-keeping effect of antimicrobial peptides and whether the form of microspheres can maintain the fresh-keeping effect for a longer time. The weight loss and firmness measurement were used to characterize the macroscopic properties of mushrooms. Changes were detected in the internal components of mushrooms, including their chemical properties and enzymes.

## 2. Materials and Methods

### 2.1. Materials

*Tricholoma matsutake*, as wild edible mushrooms, were harvested from a local grove in Kunming (Kunming, China). The mushrooms were transported to the laboratory immediately after harvest. Antibacterial peptides included cathelicidin-BF-30 (Sequence: KFFRKLKKSVKKRAKEFFKKPRVIGVSIPF, W = 3638.57 Da, peptide purity = 95.74%), which was purchased from China Peptides Co., Ltd. (ShangHai, China). Cathelicidin-BF-30 was found to exert broad antimicrobial activity against bacteria and to exhibit excellent inhibitory activity [31]. Wang [32] reported that cathelicidin-BF-30 could be an excellent therapeutic agent for acne vulgaris and that cathelicidin-BF-30 is harmless to the human body. Li [33] also showed that AMS had a proliferation effect on the cells and that the microspheres were not toxic.

### 2.2. The Preparation of the Antimicrobial Peptide Microspheres

The method for making the antimicrobial peptide microspheres was an emulsified curing method. First, 0.5 g poly(lactic-co-glycolic acid) (PLGA; 75:25) was added into 5 mL dichloromethane (DCM), and then, 50 mg of Tween80 was also added in an ice bath at a rate of 0.5 mL/min with rapid dispersion of 1 min. To obtain the colostrum, the material quickly was transferred into the solution with 20 mL polyvinyl alcohol (PVA, 5% g/mL) and continued rapid dispersion for 4 min. The emulsion with a 5% isopropyl alcohol solution (400 mL) was mixed at low speed for 3 h. At 7000 RPM and 4 °C, the mixture was centrifuged for 10 min; afterwards, the microspheres were collected and washed three times with pure water, and then, they were employed in rapid lyophilization at 55 °C.

### 2.3. Scanning Electron Microscopy Measurements

Scanning electron microscopy (Hitachi, S-3000 N, Tokyo, Japan) was used to analyze the microsphere morphology. A small amount of solution was collected before centrifugation and diluted with pure water, and the particle size distribution was measured with a laser particle size analyzer. The average value of each sample was measured 3 times.

### 2.4. Pretreatment and Preservation of Mushrooms

Each sample was sorted according to the shape and maturity of the mushrooms. Damaged or rotten mushrooms were removed. Then, the *Tricholoma matsutake* mushrooms were carefully cleaned by hand to remove mud and pine needles from the surface. The mushrooms were randomly divided into four different treatments. They were dipped into different preservatives (1.0 g/L and 0.5 g/L AMS and 1.0 g/L AMPs for 2 min at 20 °C) with a dip in distilled water used as a control. A fan generating a low-speed air flow was used to dry the *Tricholoma matsutake* mushrooms. The *Tricholoma matsutake* mushrooms were placed on a crisper (115 × 115 × 45) that was constructed from low-density polyethylene (LDPE) film, which was punched with four holes (Φ = 7 mm) and stored at 4 ± 1 °C for 16 days. The quality of the mushrooms was determined initially and after 4, 8, 12, 16, and 20 days. Three replicates from each group were randomly selected and sampled as described below.

### 2.5. Weight Loss

Weight loss was determined by the weight of the mushroom before and after storage. It was expressed as the percentage of weight loss with respect to the initial weight. Weight loss was determined gravimetrically.

### 2.6. Firmness Measurement

The firmness of the mushrooms was measured with a texture analyzer (TA-XT, Stable Micro System Ltd., London, UK) using a 2 mm-diameter cylindrical probe. Samples were penetrated to a depth of 5 mm. The speed of the probe was 2.00 mm s^−1^ during both the pretest and penetration. From the force versus time curves, the firmness was defined as the maximum force (Newton N). There were, on average, 10 mushrooms in each package.

### 2.7. Analysis of Chemical Properties

The total sugars in the mushrooms were determined according to the methods described by Miller [34] and Dubois et al. [35]. The determination of total ascorbic acid was performed as described by Hanson et al. [36] based on coupling 2,4-dinitrophenylhydrazine (DNPH) with the ketonic groups of dehydroascorbic acid through the oxidation of ascorbic acid by 2,6-dichlorophenolindophenol (DCIP), providing a yellow/orange color in acidic conditions. Mushroom tissues (10 g) were blended with 80 mL of 5% meta-phosphoric acid in a homogenizer and centrifuged. After centrifuging, 2 mL of the supernatant were poured into a 20 mL test tube containing 0.1 mL of 0.2% 2,6-DCIP sodium salt in water, 2 mL of 2% thiourea in 5% meta-phosphoric acid, and 1 mL of 4% 2,4-DNPH in 9 N sulfuric acid. The mixtures were kept in a water bath at 37 °C for 3 h and then placed in an ice bath for 10 min. Then, 5 mL of 85% sulfuric acid was added, and the mixtures were kept at room temperature for 30 min before being read at 520 nm.

### 2.8. Enzyme Assays

To analyze the enzyme activities, mushroom tissues (5.0 g) were removed using a sharp knife and then homogenized with 20 mL of 0.5 M sodium phosphate buffer (pH 6.5) containing 20 g/L of polyvinylpyrrolidone to restrict oxidation in the samples. After centrifugation for 20 min at 8000 rpm and 4 °C, the supernatant was collected and used as a crude enzyme extract for the polyphenol oxidase (PPO) and peroxidase (POD) assays.

The PPO activity was assessed using the oxidation of p-phenylendiamine by catechol as a substrate [37]. PPO was measured immediately after the extraction to avoid degradation of the enzymes. The absorbance was measured at 420 nm with a UV-vis spectrophotometer (T90, Beijing Purkinje General Instrument Co. Ltd., Beijing, China). One unit of PPO activity was defined as the amount of enzyme that caused a 0.001 increase in absorbance per minute in 1 mL in the reaction mixture.

The POD activity was measured based on the spectrophotometric measurement by using a modified method based on those described by Gao [38]. The reaction mixture for the determination of POD activity consisted of 50 mM sodium phosphate buffer (pH 6.0), 5 mM guaiacol, 5 mM H_2_O_2_, and 50 μL of tissue extract. One unit of POD activity was defined as the amount of enzyme that caused a change of 0.01 per minute at an absorbance at 470 nm under the specified conditions. The specific PPO and POD activities were expressed as Umg^−1^ protein. Measurements were replicated three times. The PPO and POD activities of the samples were calculated using the following formula:$$E=\frac{\Delta OD\times V_{T}}{0.01\times V_{1}\times t\times F_{W}}$$

$V_{T}$: The total volume of the sample solution (mL).

$V_{1}$: Test sample volume (mL).

$F_{W}:$ Weight of the sample (g).

### 2.9. Microbiological Analysis

Methods for the determination of microorganisms were reported by Simón and Gonzalez-Fandos [39]. All samples were analyzed for the mesophilic and psychrophilic bacteria counts. Twenty-five grams of mushrooms was removed aseptically from the package, weighed, and homogenized in a sterile stomacher bag for 2 min, with 225 mL of 0.1% peptone water. Further decimal dilutions were made with the same diluent. Aerobic counts were determined on plate count agar. The plates were incubated at 37 °C for 2 days for mesophilic bacteria and at 4 °C for 7 days for psychrophilic bacteria.

### 2.10. Statistical Analysis

All measurements were replicated three times, and the mean values ± standard deviations were reported for each case. An analysis of variance (ANOVA) was performed using a SPSS statistical computer software package (SPSS version 13.0). The significance of differences between mean values was assessed using the Duncan’s multiple range test at a significance level of *p* < 0.05.

## 3. Results and Discussion

### 3.1. The Morphology and Particle Size of Antimicrobial Peptide Microspheres

The experimental microsphere production rate was 78.6%, which was close to 80%, and the yield can be considered relatively high. Figure 1 shows the SEM images of freeze-dried microspheres; the microsphere surface was smooth, there were fewer adhesions, and the particle size distribution was not uniform, but this is typical of a complex emulsion/solvent extraction of microspheres. Only through the membrane emulsification method could the microsphere particle size be made uniform.

The particle size distribution range was 0.5–3 µm, and the polydispersity was 0.239 (Figure 2). Most of the particle size distribution was in the range of 800–1400 nm. A monodisperse system can make the dose more reliable and more accurate. The carrier of a particle size less than 1 micron can be avoided by capillary blockage and filtration, which can be effectively absorbed by the cells [40]. Such a particle size distribution is suitable for drug delivery systems [41].

### 3.2. Weight Loss

All samples lost weight over the 20-day storage period (Figure 3). This loss increased as the storage period progressed for all treatments. After 20 days of storage, the mushrooms that were coated with AMS had 2.72% and 2.93% weight loss, respectively, compared with the 3.34% and 3.25% weight loss values observed in the control and antimicrobial peptide groups. The weight losses were all less than 3.5% during storage. However, after 4 d of storage, the weight loss was relatively higher for the control samples than for those with an antimicrobial peptide microsphere coating (0.5 g/L and 1 g/L). Mushroom weight loss is mainly caused by water transpiration and CO_2_ loss during respiration. The thin skin of shiitake mushrooms makes them susceptible to rapid water loss, resulting in shriveling and deterioration [42]. Mushrooms can endure surface shrinkage. Additionally, there can be fresh mushroom loss due to transpiration, which decreases the weight and quality and affects the normal physiological processes, as occurs with increased enzyme activity and respiration intensity with decreased resistance to disease.

### 3.3. Firmness Measurement

Change in firmness is considered one of the chief problems in the postharvest deterioration of mushrooms [43]. Mushrooms suffer a rapid loss of firmness during senescence, which strongly contributes to its short postharvest life and susceptibility to fungal contamination. The texture of *Tricholoma matsutake* is often the first of many quality attributes that the consumer judges; it is extremely important in overall product acceptance. Mushrooms suffer a rapid loss of firmness during senescence, which strongly contributes to its short postharvest life and susceptibility to fungal contamination. From a quality perspective, the texture of *Tricholoma matsutake* is an important parameter that the consumer judges. At harvest, mushrooms are firm, crisp, and tender. However, they soften during postharvest deterioration [44]. During the process of storage, the preservative effect on *Tricholoma matsutake* hardness is shown in Figure 4. At the start of storage, the firmness increases slightly, which is likely from mushroom water loss. Subsequently, the hardness begins to decline, and softening can occur because of the degradation of cell walls in postharvest mushrooms in response to bacterial enzymes and the increased activity of endogenous autolysins [45]. Microorganisms, such as pseudomonas, degrade mushrooms by breaking down the intracellular matrix and reducing the central vacuole, resulting in partially collapsed cells and a loss of turgor. This type of bacterial-induced softening was observed in control samples, but it was inhibited by antimicrobial peptide coating treatments. The maintenance of firmness in the mushrooms that were treated with antimicrobial peptide coatings could be due to their higher antifungal activity or to the covering of the cuticle and lenticels, which reduced infection, respiration, and other senescence processes during storage. The loss of moisture during storage was responsible for firmness loss. With respect to firmness on the eighth day of storage, the control group and PLA-1.0 group had obvious reduced firmness compared with the other three groups. In 20 days, the hardness of the mushrooms with AMS-1.0 as a preservative decreased more slowly than did that of the other four groups; therefore, AMS-1.0 can be very effective in slowing the process of mushrooms becoming soft.

### 3.4. Analysis of Chemical Properties

As one of the nutritional indexes, total sugar can reflect the process of storage and quality changes in *Tricholoma matsutake*. Total soluble sugars are not only the main photosynthates in higher plants, but they are also the main form of carbohydrate metabolism and temporary storage. Five freshness modes of *Tricholoma matsutake* storage in the process of changing the total sugar content are analyzed. With early storage, the total sugar content increased (Figure 5). The increase in total sugar content in the observed samples may be due to the high respiration rate and maturation of mushrooms during storage, and it is consistent with the existing research results [15]. The total sugar levels in the antimicrobial peptide microsphere-coated (1.0 g/L) samples increased at a significantly different rate (*p* < 0.05) compared with the other three groups over the first 12 d period, resulting in a 109.8% higher than initial concentration at the termination of the experiment. The highest total sugar content was 16.54 mg/g. Total sugar began to decline in the control group and PLGA-1.0 group on the eighth day, whereas the total sugar content in the remaining three groups decreased. During storage, the continuous respiration of Tricholoma matsutake consumed microbial nutrients, resulting in the decrease in total sugar content. In the period from day 12 to day 20, the total sugar contents of the control group and PLGA-1.0 group were significantly lower than those of the other three groups.

Figure 6 shows the ascorbic acid level changes in *Tricholoma matsutake* after 20 days of storage. Even for *Tricholoma matsutake* mushrooms that were treated with both AMPs and AMS, the ascorbic acid levels decreased. The starting value for ascorbic acid was 42.9 mg/Kg, and it significantly decreased by the fourth day. On the eighth day, the control and PLGA-1.0 groups had somewhat lower ascorbic acid levels, although these levels were not significantly lower than those of the other three groups. These findings were in agreement with a study by Ayranci and Tunc [46]. Storage temperature was the main factor affecting the fluctuation of ascorbic acid content, because the temperature directly affects the enzyme activity. Keeping the appropriate low temperature in the storage environment of Tricholoma matsutake was conducive to the preservation of ascorbic acid.

### 3.5. Enzyme Assays

It is generally accepted that browning is due to the oxidation of phenolics and is caused by PPO and POD, resulting in the formation of brown-colored substances. This may be due to the oxidation of edible fungus polyphenols in the body into quinone substances, which form through a series of reactions, i.e., the Browning reactions. Figure 7a shows the influence of different preservative agents on the *Tricholoma matsutake* PPO activity. With an increase in storage time, the preservation of *Tricholoma matsutake* PPO activity had a unimodal curve change in this way. The different preservative agents for *Tricholoma matsutake* had different effects on the PPO activity. The PPO activity of both the control group and PLGA-1.0 group in the first eight days increased more rapidly, peaking within 12 days, whereas the other three groups peaked on the 16th day. The hydrogen POD and PPO activity had a similar trend as the storage time increased. Within 12 days, the POD activity of the four groups peaked (Figure 7b). When the concentration of AMS was 1.0 g/L for *Tricholoma matsutake*, the PPO activity and POD activity were significantly lower than those observed in the other three groups. It was responsible for the oxidation of phenolic compounds and the formation of brown-colored melanins, preventing the formation of brown patches and improving the appearance and color of the mushrooms [47]. At the same time, the AMS could improve the effect of the enzyme. These results suggested that the AMS coating treatment was more effective in retarding mushroom sensory deterioration.

### 3.6. Microbiological Analysis

The changes in the total aerobic bacteria counts for *Tricholoma matsutake* during storage are shown in Figure 8. In all groups, the total levels of mesophilic and psychrophilic bacteria counts steadily increased throughout the storage period. Mesophilic and psychrophilic bacteria predominated during storage in all of the analyzed samples. At harvest, the mushrooms had 1.51 and 3.94 log_10_cfug^−1^ for psychrophilic and mesophilic bacteria, respectively. These levels were higher than those reported by Cliffe-Byrnes and O’Beirne [48] for water-rinsed white button mushrooms and by Jiang et al. [49] for shiitake mushrooms. This was probably because *Tricholoma matsutake* mushrooms were harvested from a local grove and mud, and many pine needles had adhered to the stems and caps of the wild mushrooms. Following 20 days of cold storage, the microbial counts in the antimicrobial peptide and antimicrobial peptide microsphere treatment groups were increased throughout the storage period. However, their growth was slower that of the controls. AMS coating was more effective in reducing the microbial counts than other treatments. For all groups, the psychrophilic bacteria counts increased by less than two orders throughout the storage period. Mushrooms from the control group and PLGA-1.0 group had tiny brown spots on day 12 that developed into dark zones, which are characteristic of pseudomonas spoilage, by day 16. The mushrooms were highly decayed at this point, and the end of their shelf life was due to microbial spoilage. There were no differences between the AMS and the AMPs in their response to psychrophilic bacteria, but with a high concentration of AMS solution, the antimicrobial peptide preservation effect was much better for mesophilic bacteria. However, overall, the coating and control groups showed obvious differences in the number of psychrophilic bacteria. Thus, antimicrobial peptides might have stronger effects on psychrophilic bacteria.

### 3.7. Sensorial Analysis

The average values for the sensory attributes are listed in Table 1. All of the selected sensory attributes gradually decreased as the storage period advanced. This supported the validity of the chosen descriptors as indicators of mushroom deterioration.

There was no significant (*p* > 0.05) difference among the groups on day 0 of storage. The application of AMS did not influence the sensorial analysis of the samples compared with the PLGA-1.0 group and control group. A similar observation with no significant difference between the samples and controls was observed on day 4 of storage. The off-odor of mushrooms coated with AMS-1.0 and AMS-0.5 was significantly (*p* < 0.05) less than those coated with AMPs-1.0 after 16 days of storage. The growth of microorganisms may accelerate the decay of the mushroom. This phenomenon and the trend of microbial quantity was the same.

The appearance of mushrooms coated with antimicrobial peptides was significantly (*p* < 0.05) better than the control group and PLGA-1.0 group after 8 days of storage. Mushrooms in the control group and PLGA-1.0 group became unacceptable on day 16. The general acceptability of AMP-treated mushrooms fell below the limit of marketability at day 16. However, a better trend was observed for AMS-0.5 and AMS-1.0-treated mushrooms. AMS-0.5 and AMS-1.0-treated mushrooms were still acceptable in a marketable condition and received a score of 6 at the end of the storage time. The results suggested that the AMS-0.5 and AMS-1.0 treatments were more effective in retarding the deterioration of the general acceptability of mushrooms compared with the other test samples.

Fresh Tricholoma matsutake can be stored in the refrigerator freezer for 3–5 days. The experimental results showed that after fresh-keeping treatment, it was still acceptable for 8 days. To some extent, the purpose of prolonging the shelf life was achieved.

## 4. Conclusions

In this study, novel AMPs and AMS coating materials were applied for the preservation of *Tricholoma matsutake* mushrooms in storage at 4 ± 1 °C. The AMPs and AMS coatings had beneficial effects on the physicochemical and physiological quality of the mushrooms compared to the control treatment. The AMS coating had a significant effect on the texture, PPO activity, POD activity, total sugar, and microbiological analysis of *Tricholoma matsutake* wild edible mushrooms, although its effect on weight loss and ascorbic acid was limited. The result suggested that AMS coatings maintain the firmness of mushrooms and improve the postharvest quality during cold storage. These findings also suggested that AMS are promising as an edible coating for use in commercial postharvest applications to prolong the storage life of button mushrooms. Therefore, the AMS coating may provide an attractive alternative to improve the preservation qualities of *Tricholoma matsutake* during extended storage. The existing research results show that the long-term preservation of wild fungi in the form of microspheres has application prospects in order to improve the shelf-life of Tricholoma matsutake and other expensive mushroom varieties.

## Figures and Tables

**Figure 1 polymers-14-00208-f001:**
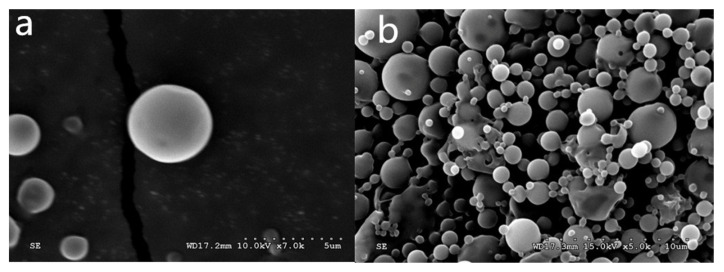
The antimicrobial peptide-PLGA microspheres SEM results; (**a**,**b**) correspond to different amplification lyophilized microspheres, multiples of (**a**) ×7000, (**b**) ×5000.

**Figure 2 polymers-14-00208-f002:**
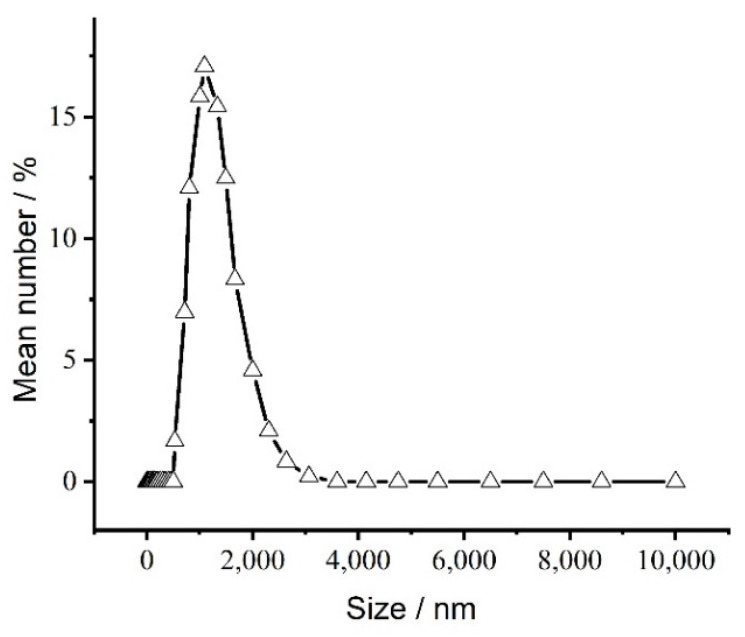
Particle size distribution of the freeze-dried microspheres.

**Figure 3 polymers-14-00208-f003:**
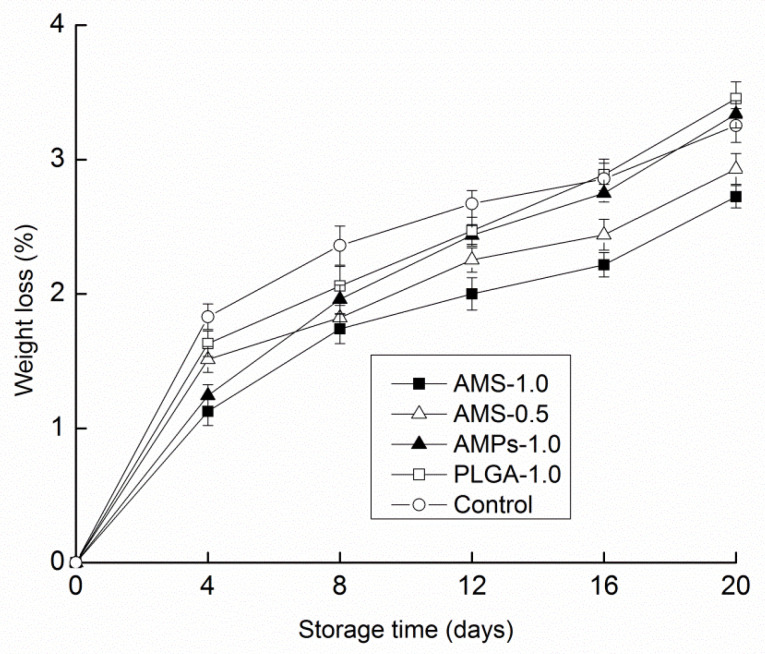
Effect of different antimicrobial coating on weight loss changes of *Tricholoma matsutake* stored at 4 ± 1 °C for 20 days. Each data point is the mean of three replicate samples.

**Figure 4 polymers-14-00208-f004:**
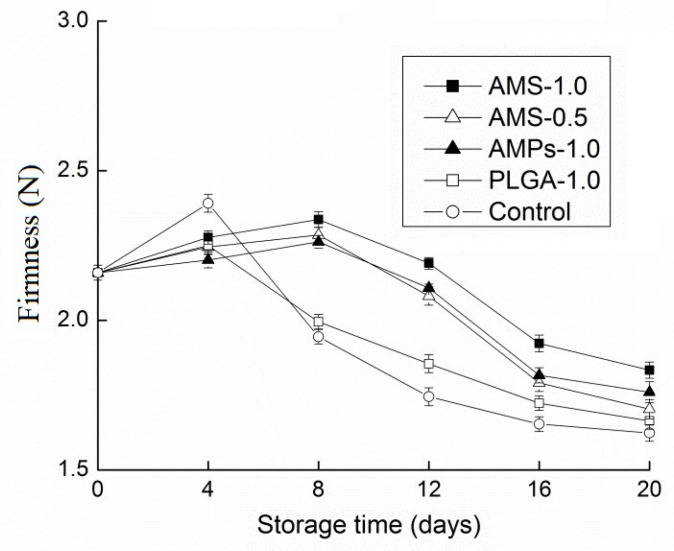
Effect of different antimicrobial coating on texture changes of *Tricholoma matsutake* stored at 4 ± 1 °C for 20 days. Each data point is the mean of three replicate samples.

**Figure 5 polymers-14-00208-f005:**
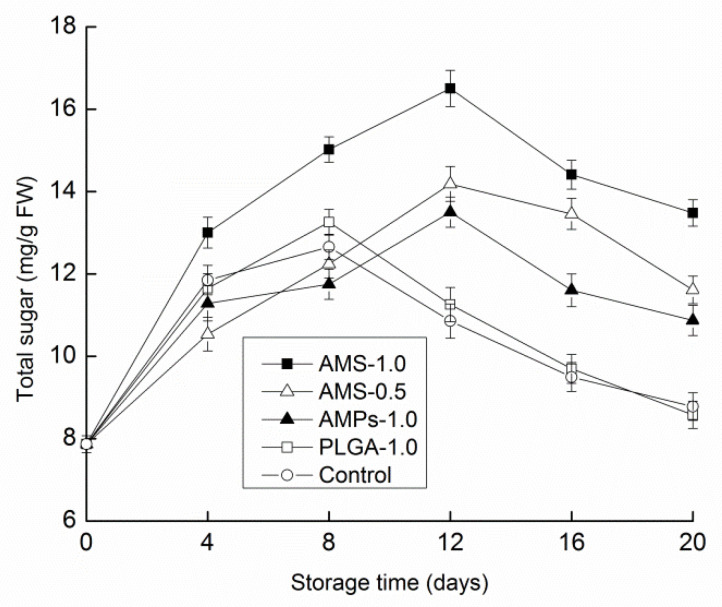
Effect of different antimicrobial coatings on total sugar changes of *Tricholoma matsutake* stored at 4 ± 1 °C for 20 days. Each data point is the mean of three replicate samples.

**Figure 6 polymers-14-00208-f006:**
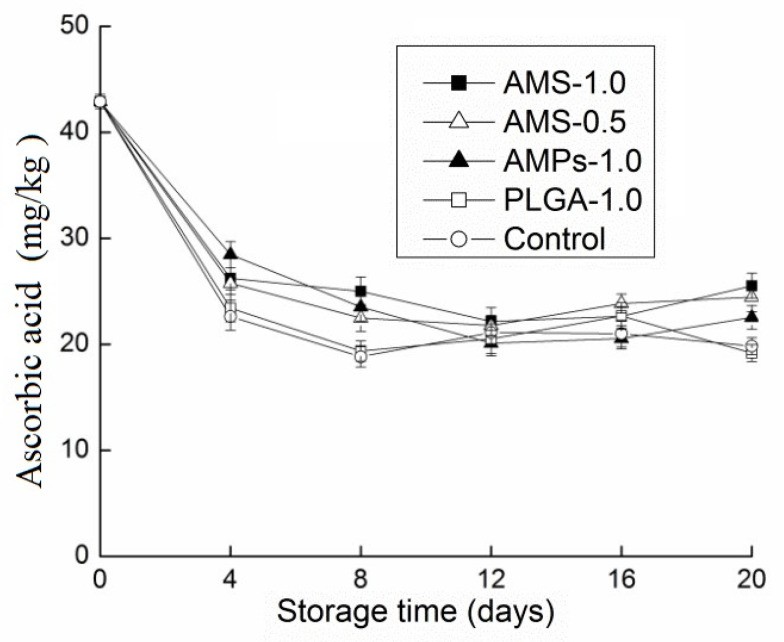
Effect of different antimicrobial coatings on ascorbic acid changes of *Tricholoma matsutake* stored at 4 ± 1 °C for 20 days. Each data point is the mean of three replicate samples.

**Figure 7 polymers-14-00208-f007:**
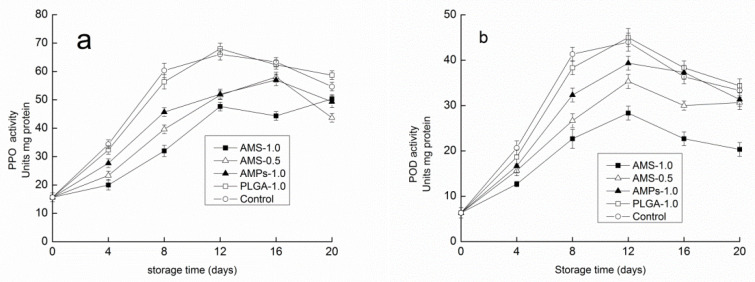
Effect of different antimicrobial coatings on PPO activity (**a**) and POD activity (**b**) changes of *Tricholoma matsutake* stored at 4 ± 1 °C for 20 days. Each data point is the mean of three replicate samples.

**Figure 8 polymers-14-00208-f008:**
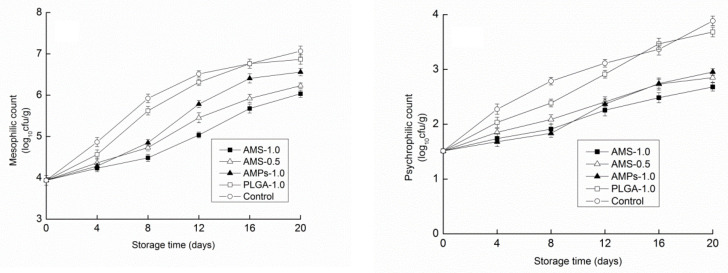
Effect of different antimicrobial coatings on mesophilic and psychrophilic count changes of *Tricholoma matsutake* stored at 4 ± 1 °C for 20 days.

**Table 1 polymers-14-00208-t001:** Effect of different treatments on the color of *Tricholoma matsutake* mushrooms stored at 4 ± 1 °C for 20 days.

Treatments	Appearance	Spoilage	Odor
0 day			
	9.93 ± 0.54	9.76 ± 0.47	10.0
4 days			
AMS-1.0	9.24 ± 0.47 ^a^	9.42 ± 0.16 ^a^	9.83 ± 0.19 ^a^
AMS-0.5	9.32 ± 0.62 ^a^	9.45 ± 0.53 ^a^	9.82 ± 0.25 ^a^
AMPS-1.0	9.20 ± 0.25 ^a^	9.51 ± 0.67 ^a^	9.89 ± 0.22 ^a^
PLGA-1.0	9.12 ± 0.16 ^a^	9.46 ± 0.25 ^a^	9.85 ± 0.31 ^a^
Control	9.07 ± 0.17 ^a^	9.43 ± 0.55 ^a^	9.74 ± 0.72 ^a^
8 days			
AMS-1.0	8.63 ± 0.83 ^a^	8.12 ± 0.82 ^a^	8.79 ± 0.56 ^a^
AMS-0.5	8.51 ± 0.74 ^a^	8.07 ± 0.14 ^a^	8.83 ± 0.49 ^a^
AMPS-1.0	8.74 ± 0.42 ^a^	8.19 ± 0.26 ^a^	8.57 ± 0.31 ^a^
PLGA-1.0	7.72 ± 0.37 ^b^	7.45 ± 0.63 ^b^	8.31 ± 0.31 ^b^
Control	7.46 ± 0.26 ^b^	7.24 ± 0.37 ^b^	8.25 ± 0.89 ^a^
12 days			
AMS-1.0	8.04 ± 0.33 ^a^	7.24 ± 0.53 ^a^	8.05 ± 0.63 ^a^
AMS-0.5	7.87 ± 0.59 ^a^	7.16 ± 0.33 ^a^	8.13 ± 0.46 ^a^
AMPS-1.0	7.93 ± 0.35 ^a^	7.09 ± 0.74 ^a^	7.24 ± 0.53 ^a^
PLGA-1.0	6.52 ± 0.14 ^b^	6.33 ± 0.44 ^b^	6.57 ± 0.26 ^b^
Control	6.05 ± 0.63 ^b^	6.04 ± 0.92 ^b^	6.43 ± 0.48 ^b^
16 days			
AMS-1.0	7.33 ± 0.52 ^a^	6.88 ± 0.74 ^a^	7.67 ± 0.27 ^a^
AMS-0.5	7.04 ± 0.71 ^ab^	6.53 ± 0.81 ^ab^	7.25 ± 0.48 ^ab^
AMPS-1.0	6.82 ± 0.35 ^b^	6.24 ± 0.42 ^b^	6.53 ± 0.31 ^b^
PLGA-1.0	5.02 ± 0.37 ^c^	5.13 ± 0.05 ^c^	4.93 ± 0.11 ^c^
Control	4.87 ± 0.83 ^c^	4.82 ± 0.22 ^c^	4.86 ± 0.61 ^c^
20 days			
AMS-1.0	6.81 ± 0.56 ^a^	6.32 ± 0.63 ^a^	6.83 ± 0.44 ^a^
AMS-0.5	6.13 ± 0.68 ^ab^	6.14 ± 0.58 ^a^	6.47 ± 0.49 ^a^
AMPS-1.0	5.67 ± 0.14 ^b^	5.75 ± 0.15 ^a^	5.52 ± 0.58 ^b^
PLGA-1.0	3.24 ± 0.72 ^c^	3.57 ± 0.63 ^c^	3.38 ± 059 ^c^
Control	3.13 ± 0.32 ^c^	3.66 ± 0.44 ^b^	3.14 ± 0.27 ^c^

^a–c^ Values followed by different letters in the same column are significantly different (*p* < 0.05), where a is the lowest value. Data are presented as mean ± standard deviation (*n* = 3).

## Data Availability

The data presented in this study are available on request from the corresponding author.

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
