# Peer review of "Effect of Antibacterial Peptide Microsphere Coating on the Microbial and Physicochemical Characteristics of Tricholoma matsutake during Cold Storage"

_polymers, 2022, doi:10.3390/polym14010208_

Round 1

Reviewer 1 Report

Li et al. presented a well-crafted and interesting manuscript about the application of enclosed antimicrobial peptides for the preservation of mushrooms. The manuscript is appropriate for Polymers and should be accepted for publication after a minor revision by the authors. My comments follow. 

Line 34 revise “valuable species”

Line 40 provide reference

Line 46 revise “severe”, define “less damaging”

Lines 54-61 revise to avoid repetition

Line 67 While technically true, I would avoid the use of the terminology “antibiotics” for AMP

Lines 67-70 revise for clearance of meaning

Lines 189-193 this part belongs to materials and methods

Line 249 In my opinion the loss of moisture during storage is responsible for firmness loss rather.

Lines 281-289 provide a plausible physiological explanation about the fluctuation of ascorbic acid during storage.

Line 331 we cannot refer to contamination level here as the mushrooms were already contemned before the experimental observations

Author Response

Dear Reviewer:
Thank you for your letter and for the reviewer’s comments concerning our manuscript entitled “Effect of antibacterial peptide microsphere coating on the microbial and physicochemical characteristics of Tricholoma matsutake during cold storage” (ID: 1513704). Those comments are all valuable and very helpful for revising and improving our paper, as well as the important guiding significance to our researches. We have studied comments carefully and have made correction which we hope meet with approval. Revised portion are marked in red in the paper. The main corrections in the paper and the responds to the reviewer’s comments are as flowing:
Responds to the reviewer’s comments:

  1. Response to comment: Line 34 revise “valuable species”

Response: In line34, the most valuable species has been replaced with more precious and relatively expensive species.

  1. Response to comment: Line 40 provide reference

Response: In this article of “Evaluation of biodegradable film packaging to improve the shelf-life of Boletus edulis wild edible mushrooms”, it was already mentioned that the high respiration rate, lack of physical protection to avoid water loss and changes from microbial attack are often associated with a rapid decrease in mushroom quality. And this reference has been added here.

  1. Han L.L., Qin Y.Y., Liu D., Chen H.Y., Li H.L., Yuan M.L.,2015. Evaluation of biodegradable film packaging to improve the shelf-life of Boletus edulis wild edible mushrooms. Innov. Food Sci. Emerg., 29, 288-294.

  1. Response to comment: Line 46 revise “severe”, define “less damaging”

Response: We have re-written this part according to the Reviewer’s suggestion. There is a general trend in mushroom preservation research towards the development of preservation techniques that are practical and operable in order to reduce damaging to food products. These damages include mushroom loss of quality, contributing to their deterioration through browning, cap opening, stipe elongation, cap diameter increase, weight loss and texture changes [13].

  1. Response to comment: Lines 54-61 revise to avoid repetition

Response: We have re-written this part according to the Reviewer’s suggestion. The statements of “They have been shown to protect products from microbial and mechanical damage, provide an aesthetic appearance, and prevent the escape of volatile matters [19]. Edible coatings are based on biodegradable and edible materials from natural sources, and they therefore satisfy environmental concerns and are responses to customer demands [20]. Coating preservation technology has focused on surface coating technology and the different types of coatings that are used to protect food from the effects of environmental factors, such as light, oxygen and microorganisms [21]” have been revised.

  1. Response to comment: Line 67 While technically true, I would avoid the use of the terminology “antibiotics” for AMP

Response: The AMPs is an abbreviation of antimicrobial peptides, which is written in many literatures on antimicrobial peptides.

[1] Xueqing Xu, Ren Lai. The Chemistry and Biological Activities of Peptides from Amphibian Skin Secretions. Chemical Reviews, 2015, 115(4):1760-1846

[2] Md Abdul Hakim, Shilong Yang, Ren Lai. Centipede Venoms and Their Components: Resources for Potential Therapeutic Applications. Toxins, 2015, 7, 4832-4851

  1. Response to comment: Lines 67-70 revise for clearance of meaning

Response: We have re-written this part according to the Reviewer’s suggestion.

In recent years, antimicrobial peptides (AMPs) as potential new antibiotics to solve the problem of resistance, AMP as a kind of biological epidemic prevention system against exogenous pathogens of natural antibacterial peptides which are small molecule composed of 12-100 amino acids [24, 25].

  1. Response to comment: Lines 189-193 this part belongs to materials and methods

Response: We are very sorry for our negligence of this part. And the corresponding content has been modified to the experimental operation part.

  1. Response to comment: Line 249 In my opinion the loss of moisture during storage is responsible for firmness loss rather.

Response: We have made correction according to the Reviewer’s comments.

  1. Response to comment: Lines 281-289 provide a plausible physiological explanation about the fluctuation of ascorbic acid during storage.

Response: Storage temperature was the main factor affecting the fluctuation of ascorbic acid content, because the temperature directly affects the enzyme activity. Keeping the appropriate low temperature in the storage environment of Tricholoma matsutake was conducive to the preservation of ascorbic acid.

  1. Response to comment: Line 331 we cannot refer to contamination level here as the mushrooms were already contemned before the experimental observations

Response: As Reviewer suggested that the line 331 has been deleted.

Special thanks to you for your good comments.

Reviewer 2 Report

The submitted article presents an interesting idea of using antibacterial peptide microsphere coating for antibacterial purposes during the cold storage of Tricholoma matsutake. However, some issues must be enhanced to increase article quality and relevance. Therefore, please consider reviewing the following issues, incorporating new information into the original article: 

  1. Lines 13, 84,106 and 192: Avoid using "we". Please, revise the rest of the article. Also, check for phrases like "Our study.." (line 18).
  2. Please enhance the last paragraph of the introduction, considering declaring the main target of research and novelty.
  3. Lines 101-113: Please consider rewriting the paragraph, it is challenging to read, and the redaction needs to be revised. In addition, characteristics of mentioned equipment must be included.
  4. This study referenced some methods without further explanation than incorporating reference. Please consider including more information to enhance the methods section. Line 148. Please explain the measurement method.
  5. Figures 2, 4, 6, 7, and 8. Please improve the resolution and revise the font type.
  6. Line 195. The SEM should be Hitachi.
  7. Please consider using the same scale and SEM configuration for both pictures in Figure 1. It will improve comprehension of the reported pore size distribution.
  8. Please consider including previously reported values of total sugar content in section 3.4.
  9. To avoid misunderstandings, please consider improving the redaction of these sentences, incorporating a storage time allusion:
  • Lines 265-266: The observed increase in total sugar content of the samples might be due to the high respiration rate and ripening of the mushrooms during storage…
  • Lines 272-274: The total sugar content decreased because Tricholoma matsutake continued to respire during storage, consuming and reducing the microbial nutrients….
  1. A paragraph is expected after section 3.7. Please rearrange Table 1 position in the article.
  2. Please consider including more information to justify the research novelty statement in the conclusions.

Author Response

Dear Reviewer:

Thank you for your letter and for the reviewer’s comments concerning our manuscript entitled “Effect of antibacterial peptide microsphere coating on the microbial and physicochemical characteristics of Tricholoma matsutake during cold storage” (ID: 1513704). Those comments are all valuable and very helpful for revising and improving our paper, as well as the important guiding significance to our researches. We have studied comments carefully and have made correction which we hope meet with approval. Revised portion are marked in red in the paper. The main corrections in the paper and the responds to the reviewer’s comments are as flowing:

1 Response to comment: Lines 13, 84,106 and 192: Avoid using "we". Please, revise the rest of the article. Also, check for phrases like "Our study.." (line 18).

Response: We have re-written these parts according to your suggestion to revise the first-person usage in the whole manuscript.

  1. Response to comment: Please enhance the last paragraph of the introduction, considering declaring the main target of research and novelty.

Response: As Reviewer suggested that the corresponding content is added to the manuscript. In this work, the antimicrobial peptide microspheres and antimicrobial peptide as preservatives were carried out to the preservation experiments of Tricholoma matsutake. The purpose is to test the fresh-keeping effect of antimicrobial peptides and whether the form of microspheres can maintain the fresh-keeping effect for a longer time. The weight loss and firmness measurement were used to characterize the macroscopic properties of mushrooms. Chemical properties and enzyme were detected changes in the internal components of mushrooms.

  1. Response to comment: Lines 101-113: Please consider rewriting the paragraph, it is challenging to read, and the redaction needs to be revised. In addition, characteristics of mentioned equipment must be included.

Response: The paragraph has been rewritten. The description of sample preparation and sample detection methods has been separated. The details are marked in red in the manuscript.

  1. Response to comment: This study referenced some methods without further explanation than incorporating reference. Please consider including more information to enhance the methods section. Line 148. Please explain the measurement method.

Response: The total sugars in the mushrooms were determined according to the methods described by Miller and Dubois’s experimental research methods. The determination of total ascorbic acid was performed as described by Hanson [36].

  1. Response to comment: Figures 2, 4, 6, 7, and 8. Please improve the resolution and revise the font type.

Response: Figures 2, 4, 6, 7, and 8 have been remapped and the image resolution has been improved. And the font type of figures 4, 6, 7, and 8 have been revised.

  1. Response to comment: Line 195. The SEM should be Hitachi.

Response: It has been revised.

  1. Response to comment: Please consider using the same scale and SEM configuration for both pictures in Figure 1. It will improve comprehension of the reported pore size distribution.

Response: In the whole manuscript, only one proportion of microspheres were prepared. In Figure 1, a is an enlarged part of figure B, in order to more clearly describe the morphology of microspheres, and figure B is a description of the whole picture of prepared microspheres.

  1. Response to comment: Please consider including previously reported values of total sugar content in section 3.4.

Response: the statement of “The highest total sugar content was 16.54 mg/g” was added.

  1. Response to comment: To avoid misunderstandings, please consider improving the redaction of these sentences, incorporating a storage time allusion:

Lines 265-266: The observed increase in total sugar content of the samples might be due to the high respiration rate and ripening of the mushrooms during storage…

Response: Lines 265-266, the statements of “The observed increase in total sugar content of the samples might be due to the high respiration rate and ripening of the mushrooms during storage…” were corrected as “The increase of total sugar content in the observed samples may be due to the high respiration rate and maturation of mushrooms during storage, and it is consistent with the existing research results [14].”

Lines 272-274: The total sugar content decreased because Tricholoma matsutake co ntinued to respire during storage, consuming and reducing the microbial nutrients….

Response: Lines 272-274 the statements were corrected as “During storage, the continuous respiration of Tricholoma matsutake consumed microbial nutrients, resulting in the decrease of total sugar content.”

  1. Response to comment: A paragraph is expected after section 3.7. Please rearrange Table 1 position in the article.

Response: The paragraph of “Fresh Tricholoma matsutake can be stored in the refrigerator freezer for 3-5 days. The experimental results showed that after fresh-keeping treatment, it was still acceptable for 8 days. To some extent, the purpose of prolonging the shelf life was achieved.” were added.

  1. Response to comment: Please consider including more information to justify the research novelty statement in the conclusions.

Response: the statements of “The existing research results show that the long-term preservation of wild fungi in the form of microspheres has application prospects in order to improve the shelf life of Tricholoma matsutake and expensive others.” were added in the conclusions.

    We tried our best to improve the manuscript and made some changes in the manuscript.  These changes will not influence the content and framework of the paper.
We appreciate for Reviewer warm work earnestly, and hope that the correction will meet with approval.
Once again, thank you very much for your comments and suggestions.

Round 2

Reviewer 2 Report

Dear Authors,

Thanks for following reviewer requests,

Best Regards

The reviewer